# Controlling the Expression Level of the Neuronal Reprogramming Factors for a Successful Reprogramming Outcome

**DOI:** 10.3390/cells13141223

**Published:** 2024-07-20

**Authors:** Natalie Mseis-Jackson, Mehek Sharma, Hedong Li

**Affiliations:** 1Department of Neuroscience & Regenerative Medicine, Medical College of Georgia, Augusta University, Augusta, GA 30912, USA; njackson4@augusta.edu; 2Department of Biological Sciences, College of Science & Mathematics, Augusta University, Augusta, GA 30912, USA; mesharma@augusta.edu

**Keywords:** neuronal reprogramming, NeuroD1, Sox2, Ngn2, Ascl1, microRNA

## Abstract

Neuronal reprogramming is a promising approach for making major advancement in regenerative medicine. Distinct from the approach of induced pluripotent stem cells, neuronal reprogramming converts non-neuronal cells to neurons without going through a primitive stem cell stage. In vivo neuronal reprogramming brings this approach to a higher level by changing the cell fate of glial cells to neurons in neural tissue through overexpressing reprogramming factors. Despite the ongoing debate over the validation and interpretation of newly generated neurons, in vivo neuronal reprogramming is still a feasible approach and has the potential to become clinical treatment with further optimization and refinement. Here, we discuss the major neuronal reprogramming factors (mostly pro-neurogenic transcription factors during development), especially the significance of their expression levels during neurogenesis and the reprogramming process focusing on NeuroD1. In the developing central nervous system, these pro-neurogenic transcription factors usually elicit distinct spatiotemporal expression patterns that are critical to their function in generating mature neurons. We argue that these dynamic expression patterns may be similarly needed in the process of reprogramming adult cells into neurons and further into mature neurons with subtype identities. We also summarize the existing approaches and propose new ones that control gene expression levels for a successful reprogramming outcome.

## 1. Introduction

Significant advancement has been made over the past decades in regenerative medicine for repairing the diseased/injured central nervous system (CNS), which has very limited regenerative capacity. One of the major milestones of such advancement is the development of induced pluripotent stem cells (iPSCs) [1,2]. While traditional stem cell therapy that involves transplantation of foreign cells faces immunorejection and ethical concerns, the promise of iPSC strategy is to generate a substantial quantity of neural cells from patients’ own skin fibroblasts, which are readily assessable and self-renewable, to replenish the lost neurons and other cell types in disease/injury conditions of the CNS [3]. In general, once iPSC lines are established, they can be either expanded and stored or differentiated into somatic cells of different lineages [4]. However, the protocols of iPSC generation are usually lengthy and require extensive culturing and colony selection.

Neuronal reprogramming is a revolutionary concept of changing the cell fate of developmentally committed non-neuronal somatic cells to neuronal phenotype by intrinsic and/or extrinsic manipulations [5,6,7,8]. In contrast to the scenario of iPSCs where somatic cells are induced to “dedifferentiate” into pluripotent stem cells, which can be subsequently differentiated into neurons, neuronal reprogramming directly generates neurons from somatic cells without going through a primitive stem cell stage. More recently, in vivo neuronal reprogramming has brought the reprogramming approach to another level [9,10]. Converting non-neuronal cells to neurons in situ in CNS tissues by overexpressing reprogramming factors avoids the detrimental effects associated with cell culturing, such as oxidative stress and DNA damage [11,12]. Thus far, neuronal reprogramming has been successfully demonstrated in numerous injury/disease models [9,13,14,15,16,17,18,19,20,21]. Somewhat dampening the enthusiasm in the field is the recent controversy over the validation of reprogrammed neurons by Neuronal differentiation 1 (NeuroD1) vs endogenous neurons [22,23]. This is arguably due to the use of the adeno-associated virus (AAV) gene delivery system that causes gene expression leakage in the endogenous neurons, especially with high titers of AAVs injected into the brain [24,25]. Although the percentage of NeuroD1-reprogrammed neurons may have been overestimated previously, neuronal reprogramming does occur with AAV-mediated gene delivery, as recently demonstrated by the two-photon imaging analysis [26]. Therefore, in vivo neuronal reprogramming is still a feasible approach and holds promise in repairing CNS disease and injury, but stringent validation methodologies, such as cell-lineage tracing, would need to be applied for convincing interpretations [27]. In this review, we discuss the major reprogramming factors, focusing on their dynamic expression levels during development and how we may apply them for a successful reprogramming outcome. We highlight the importance of controlling expression levels of the reprogramming factors during the neuronal reprogramming process, and provide potential ways to achieve that.

## 2. Reprogramming Factors during Development

The frequently applied neuronal reprogramming factors are mostly pro-neurogenic transcription factors that play key roles during CNS development. These include the *basic helix-loop-helix* (*bHLH*) gene family members *Neurogenin1/2* (*Ngn1/2*), *Achaete-scute family bHLH transcription factor 1* (*Ascl1*), and *NeuroD1*, all of which have unique spatiotemporal expression patterns during development and regulate neurogenesis in different regions of the CNS [28]. The reprogramming factor *SRY-box transcription factor 2* (*Sox2*) belongs to the high mobility group (HMG) domain-containing gene family, and is a transcription factor mainly expressed in neural progenitors [29,30]. Among others, all these reprogramming factors have been demonstrated to reprogram somatic cells to neurons, especially in in vivo settings. We believe that understanding how these factors behave during neurogenesis will aid in designing and improving strategies for the reprogramming approach. 

### 2.1. Expression Pattern and Function

Besides their expression and function in other tissues, these reprogramming factors have been thoroughly characterized in the developing CNS. Taking the embryonic neocortex as an example, Ngn1/2 are expressed mainly in the ventricular zone (VZ) of the dorsal neocortex, where they promote cell cycle exit of the neural progenitors as they migrate toward the outer layers of the neocortex [28]. NeuroD1 is a direct downstream target of Ngn1/2 and turns on its expression in the subventricular zone (SVZ), where neural progenitors differentiate to post-mitotic neurons [31]. As the newborn neurons further migrate outwards and differentiate, NeuroD1 expression diminishes, while its downstream effectors NeuroD2/4 highly express in the outer cortical layers [32]. Therefore, as development progresses in the neocortex, there are waves of pro-neurogenic transcription factors that act in sequence to coordinate neurogenesis (Figure 1A,B). Both Ngn1/2 and NeuroD1 are major cell fate determinants for glutamatergic neurons of the dorsal neocortex [33]. In contrast, Ascl1 is mainly expressed in the ventral telencephalon [34], namely the lateral and medial ganglionic eminence (LGE and MGE), where it instructs the neurogenesis of GABAergic interneurons [35] (Figure 1A,B). Ascl1 turns on the expression of Dlx2 [31], a homeobox-containing transcription factor that substantiates GABAergic interneuron phenotype [36] and facilitates their tangential migration into the dorsal neocortex [37]. In the MGE, Ascl1 and Dlx2 also interact with another bHLH transcription factor, Olig2, for the generation of oligodendrocytes [38]. Sox2 is expressed in the VZ of both dorsal and ventral telencephalon, and is important for maintaining neural progenitor properties [29,30] (Figure 1B). Therefore, the spatial expression patterns of these transcription factors also correlate with the temporal progression of neurogenesis, since cells that migrate to different layers of the neocortex represent distinct neurogenic states. 

### 2.2. Expression Regulation

The expression of these transcription factors is tightly regulated by the dorsoventral gradients of the morphogenic factors through distinct receptor signaling pathways [39]. In some cases, they also regulate each other and compete to create boundaries of distinct CNS domains. For example, the discrete expression patterns of Ngn1/2 and Ascl1 separate dorsal and ventral telencephalon. Thus, *Ngn1/2* double-knockout results in the misexpression of Ascl1 in the dorsal cortex and ectopic generation of GABAergic neurons [40]. In addition, the relative level of one transcription factor to others may determine the output of neuronal subtypes. During forebrain development, Ngn2 and NeuroD1 have been shown to compete with Ascl1 in the determination of neuronal subtypes [41]. The dynamic regulation of expression is further manifested by the interesting phenomenon of an oscillating expression level of Ngn2 in neocortical progenitors [42]. This is mainly achieved by the combination of precise transcriptional regulation and rapid protein degradation. Indeed, Ngn2 protein has been estimated to have a 30 min half-life in culture, and is subject to rapid ubiquitination and proteasome-mediated degradation [43]. Interestingly, Ngn2 oscillates out of phase with another oscillating transcription factor Hes1 in the dorsal cortical progenitors, a pattern that is important to control the timing of neurogenesis [42]. The oscillatory expression pattern of the transcription factors is unlikely to be present in the neuronal reprogramming settings, since their expression is usually under strong and constitutive promoters in the vector constructs. However, protein degradation would likely occur during the reprogramming process to affect their protein levels through mechanisms such as ubiquitination, which is largely an unexplored research area and a potential target for improving neuronal reprogramming outcome. Another level of gene expression regulation is the post-transcriptional regulation mediated by miRNAs. Although miRNAs are important gene expression regulators during neurogenesis [44], we will not discuss them here, since the 3′-untranslated regions (UTRs) of mRNAs, which contain their target sites, are usually omitted in the expression vectors for neuronal reprogramming. Therefore, expression levels of the reprogramming factors are not directly regulated by miRNAs unless their target sites are included in the vector (see below for an example). Taken together, the expression of the reprogramming factors is dynamically regulated throughout neurogenesis. Although the molecular context in adult cells may be very different from neural progenitors during development, we argue that a more sophisticated gene expression regulation, rather than simple overexpression, is warranted for improving reprogramming outcome. Constitutively high expression of these factors may be necessary for reprogramming to occur, but may also cause problems such as cell death [45,46]. 

## 3. Reprogramming Factors during Reprogramming

### 3.1. Mechanisms of Action

Reprogramming factors are usually “pioneer” transcription factors, i.e., they have the ability to access closed chromatins, recruit facilitating factors, and initiate downstream gene expression programs for a new cell identity [47,48,49,50]. These properties appear to be essential to cell reprogramming, since differentiated somatic cell types possess distinct chromatin states, with some regions being active and others silenced. As a pioneer factor, NeuroD1 reprograms the chromatin landscape to elicit neuronal programming in embryonic stem (ES) cells [51] and microglia [52] when overexpressed. Mechanistically, ectopic expression of NeuroD1 can induce the expression of endogenous NeuroD1, as well as other downstream target genes such as Hes6 and NeuroD4 [51], which may be important effectors for NeuroD1-mediated neuronal conversion. MicroRNAs (miRNAs) also play a role during NeuroD1-mediated neuronal reprogramming. Our recent research showed that miR-375 is drastically induced by NeuroD1 in human astrocyte cultures and promotes survival of the reprogrammed neurons by modulating the expression of its targets, the RNA-binding protein family genes *nELAVLs*, which are also highly induced by NeuroD1 [46]. Interestingly, miR-375 and *nELAVLs* are also upregulated by Ngn2 and Ascl1, suggesting a conserved mechanism by this interacting pair during neuronal reprogramming [46]. Cell reprogramming may share many conserved mechanisms during development, but can also assign new roles to the existing mechanisms. For example, to improve reprogramming efficiency by Sox2, a selective screen was conducted and showed that impairing the p53-p21 pathway inhibits cell cycle exit and significantly boosts the overall production of reprogrammed neuroblasts [53]. In addition, molecular characterization at single cell level has greatly contributed to our understanding of neuronal reprogramming, a unique biological process. Single-cell RNAseq not only provides mechanistic insights of the reprogramming process, but also facilitates the confirmation of neuronal identity of the reprogrammed cells [54,55].

### 3.2. Expression Level

While much effort has been given in searching for transcription factors that are capable of neuronal reprogramming, the level of expression of these factors is less determined to break the cell identity barrier and achieve neuronal reprogramming. A good example of testing expression level of the reprogramming factor is the neuronal reprogramming from microglia by the overexpression of NeuroD1. NeuroD1 has been a controversial factor for astrocyte-to-neuron (AtN) reprogramming, as mentioned above. There is also a controversy about whether NeuroD1 can convert microglia to neurons in the brain [56,57]. The Nakashima group first demonstrated that overexpression of NeuroD1 can convert microglia to a neuronal phenotype both in culture and in the brain [52]. Later, the Peng group presented contradictory data that NeuroD1 cannot convert microglia to neurons, but rather induces apoptosis in microglia, and that the “reprogrammed neurons” are in fact endogenous neurons that have been mislabeled by lentivirus infection [58]. This discrepancy could be due to the difference in the relative expression level of NeuroD1 between the two experiments. Although the conclusion is still not final, the Nakashima group did provide strong evidence claiming that the expression level of NeuroD1 is critical for neuronal reprogramming to occur [59]. In that report, they applied a tetracycline-inducible (i.e., Tet-on) promoter to quantitatively control NeuroD1 expression level, and demonstrated that doxycycline (1 µg/mL) is the minimal dose to induce significant microglia-to-neuron reprogramming in culture [59]. They also increased NeuroD1 expression level by repetitive lentivirus infection, arguing that a higher number of viral particles may enter the cells by repetitive infection, thereby increasing the overall NeuroD1 expression level per cell [59]. Regardless of whether NeuroD1 can convert microglia to neurons in the brain, these Tet-on experiments in culture are convincing in that a certain level of NeuroD1 must be reached to break down the cell identity barrier and initiate the neuronal reprogramming process. Most overexpression vectors utilize strong promoters such as universal ones (i.e., CAG [16,58]) or cell-type-specific ones (i.e., GFAP [16], CD68 [21,58], and NG2 [14,16]) to achieve a high expression level of the transgenes. However, whether this high expression level is high enough for reprogramming to occur is a question to be tested. In the case of microglia-to-neuron reprogramming, this high level of NeuroD1 may not be enough, and an even higher level by repetitive lentivirus infection is required for successful reprogramming. Consistent with this notion, a recent study supports a “binary switch” mechanism of reprogramming factors, in that cell fate conversion occurs only when the expression of the reprogramming factors reaches a critical expression threshold [60].

One thing worthy of pointing out is that repetitive virus infection may not only increase the transgene expression level by increasing the number of viral particles per cell as suggested [59], but may also change the cellular state of the infected cells. This has been demonstrated in astrocytes that high titers of AAV infection activate astrocytes and make them more like reactive astrocytes [61]. Reports have shown that reactive astrocytes acquire a gene expression profile that is more like that of stem/progenitor cells [62] and may be more easily reprogrammed to other cell types. Therefore, the expression level threshold of reprogramming factors could be different for different cell types and their activation state. Consistently, while NeuroD1 reprograms reactive astrocytes in disease/injury conditions with high efficiency [16,17], NeuroD1-mediated neuronal conversion from non-reactive astrocytes in different regions of the adult brain has been difficult to achieve [63]. Specifically, intravascularly delivered NeuroD1-expressing AAV induced only 2.42% AtN conversion in the striatum, but none in the cortex of adult mice [63].

### 3.3. Expression Duration

While a sufficient expression level of the reprogramming factors is required to break down the genomic barrier and change the cellular identity, the expression duration of these factors may also play a role in the reprogramming process. The question is how long the high expression of the reprogramming factors is required and whether it is necessary to turn off their expression after reprogramming is completed. Persistent expression of the reprogramming factors in the newly generated neurons may have unwanted effects on their maturation and integration into the existing neuronal circuitry. In the case of NeuroD1, a report has shown that transient NeuroD1 expression as achieved by a tetracycline-inducible system is sufficient to initiate neuronal programs in ES cells and generate functional mature neurons [51]. Whether a similar mechanism exists during the reprogramming process of adult cells is a question to be addressed. On the other hand, persistent reprogramming factor expression may be beneficial in generating specific neuronal subtypes of the reprogrammed neurons. For instance, NeuroD1 is a glutamatergic neuronal lineage transcription factor during neurogenesis [64]. While most mature glutamatergic neurons turn off NeuroD1 expression, the glutamatergic neurons that still maintain high levels of NeuroD1 expression even postnatally are granule neurons in the cerebellum and hippocampus [65]. Therefore, persistent NeuroD1 expression in the reprogrammed neurons from astrocytes may contribute to the fact that they are mostly of glutamatergic phenotype [16]. Sox2 is a neural stem cell transcription factor that is important for maintaining stem cell properties [29,30]. Sox2 is also a reprogramming factor that has been successfully demonstrated to reprogram multiple glial cell types into neuroblasts [15,53]. Most reprogramming studies that involve Sox2 overexpression utilize glial cell-specific promoters such as *GFAP* and *NG2* [14,15,53,66]. The cell-type-specific promoters ensure a high expression level in the target glial cells and diminished expression in the reprogrammed neurons due to the inactivity of these promoters in neurons. In fact, persistent Sox2 expression under a universal promoter restricts its reprogramming capacity [67], and may even raise the concern of potential tumorigenicity. Even with diminished Sox2 expression in neurons by glial cell-specific promoters, the acquisition of mature neuronal phenotype with Sox2-mediated reprogramming is limited unless additional maturation strategies such as neurotrophic factors are applied [14,15,53]. Therefore, the dynamic expression pattern of the reprogramming factors is an important variable in neuronal reprogramming studies, and can be optimized to improve the reprogramming outcome.

## 4. Potential Strategies to Control Expression of the Reprogramming Factors

### 4.1. Cell-Type-Specific Promoters

To target non-neuronal cells for neuronal reprogramming, cell-type-specific promoters are often applied. This approach takes advantage of the fact that certain genes are expressed in a cell-type-specific manner. The *GFAP* gene promoter is a classic promoter to drive gene expression in astrocytes [68]. Over the years, modifications have been made to increase gene expression specificity and/or reduce the length of the promoter sequence to fit into small vector systems such as AAV [69]. For example, combinations of different enhancer elements have been demonstrated to alter cell type specificity of the *GFAP* basal promoter [70,71]. Considering the limited expression pattern of GFAP in the brain, other promoters that target astrocytes for gene expression have been developed, including *GLAST* and *ADL1H1*, most of which have also been combined with the *Cre-ERT2* line [72] to add timing control with tamoxifen treatment [73,74]. By far, the most stringent astrocyte-specific gene expression is achieved with the *ADL1H1-Cre-ERT2* [74]. Another thing to keep in mind is that different AAV serotypes have some degree of preference in infection to different cell types, and that the rational engineering of AAV capsids and the development of novel synthetic AAV serotypes will further facilitate cell-type targeting in combination with the promoter-based approach [75,76,77]. Thus far, cell-type-specific promoters have been used to target microglia [52,58] and NG2 glia [14] for neuronal reprogramming with success. 

### 4.2. Drug-Controllable Promoters

To have more flexibility in controlling the timing of gene expression, numerous drug-controllable promoters have been developed [78]. Apart from the *Cre-ERT2* system mentioned above, the most used of such promoters is the tetracycline-controllable promoter [79,80], which is derived from the bacterial tetracycline resistance operon. This promoter has two versions (Tet-On and Tet-off), and depending on which version is used, it either turns on or turns off gene expression when tetracycline/doxycycline is applied. The Tet-On/Off system usually contains two components, with one being the Tet responsive element (*TRE*), and the other being the tetracycline trans-activator (tTA) or reverse tetracycline trans-activator (rtTA) that can bind the drug and regulate gene expression [81]. The tetracycline-controllable gene expression system not only can turn on and off gene expression, but can also control expression level in a dose-dependent manner [79]. Another advantage of this system is the reversibility, i.e., one can reverse the effect using drug removal. Cell type specificity of gene expression can also be achieved with the tetracycline-controllable system by incorporating cell-type-specific gene promoters so that spatiotemporal gene expression control is feasible both in vitro and in vivo [82,83]. However, the leakiness of the system (i.e., the relatively high level of basal expression in the absence of the drug) has limited its use [84], although continuous effort has been made to improve both drug sensitivity and tightness of gene expression control [85,86,87,88]. In the neuronal reprogramming field, the Tet-On/Off system is more often applied in culture experiments where purified cell types are tested. For example, the Tet-on system has been applied to initiate neuronal differentiation from ES cells and induce neuronal reprogramming from a variety of cell types, including fibroblasts [5,6], hepatocytes [89], astrocytes [55], and microglia [59], in culture, thus making it a rather universal system for direct neuronal conversion. No reports with Tet-on system have been published for in vivo neuronal reprogramming thus far.

### 4.3. MiRNA-Mediated Inhibition

MiRNAs are a class of short non-coding RNAs that regulate gene expression in a post-transcriptional manner. MiRNAs usually bind to the 3′-UTR of the target mRNAs in a sequence-dependent manner and reduce gene expression level [90,91,92,93,94,95,96]. Like the cell-type-specific genes, numerous miRNAs have been identified to show cell-type-specific expression patterns. Given the inhibitory nature of their mechanism of action, miRNAs are often incorporated in the expression constructs for a “de-targeting” strategy. For instance, miR-124 is highly expressed in mature neurons, but not in other cell types of the CNS [90,91,92,93,94,95,96]. Therefore, miR-124 target sites can be included in the expression constructs that target astrocytes using a *GFAP* promoter to further minimize any possible “leaky” expression in neurons [97,98,99]. A great example of utilizing miRNAs as a de-targeting strategy is a recent report on the development of a toolbox expression system for targeting astrocytes in different brain regions [100]. The authors engineered a cassette of four copies of six miRNA targeting sequences that suppress transgene expression specifically in neurons and endothelial cells. By this extensive de-targeting strategy combined with a *GFAP* promoter (*GfaABC1D*), the resulting vector showed the highest gene expression specificity (>99%) in astrocytes of the mouse brain, which will for certain maximize the accuracy and resolution of cell type manipulating approaches such as neuronal reprogramming. Furthermore, it has been shown that the miRNA-mediated de-targeting approach can be even more effective than promoter-mediated ones in targeting the liver for gene expression [101]. 

Another potential application of miRNA target sites is to create a dynamic expression pattern of the transgenes. In the NeuroD1-mediated neuronal reprogramming model, cells experience a change in cell type, for example, from astrocytes to neurons, with highly expressed NeuroD1. As mentioned above, miR-124 is neuron-specific and highly expressed in mature neurons. Therefore, if we integrate the miR-124 target site into the NeuroD1 expression vector, we can achieve a high level of NeuroD1 expression in astrocytes (low in miR-124) for neuronal conversion to occur, and yet a reduced level of NeuroD1 in converted neurons (high in miR-124) (Figure 2A). We believe that a continuously high level of NeuroD1 in converted neurons drives them into the glutamatergic subtype since NeuroD1 is a glutamatergic neuronal lineage transcription factor during neurogenesis [64]. Thus, reducing the NeuroD1 level in the converted neurons may allow for the generation of diversified neuronal subtypes and improved functional recovery in disease/injury models (Figure 2B). In line with this hypothesis, the expression level of NeuroD1 has been related to the preference of distinct subtypes among the newly reprogrammed neurons in the retina [102].

## 5. Additional Notes

Thus far, we have mostly discussed gene expression control in the scenario of single reprogramming factors. However, a combination of different reprogramming factors has often been applied in neuronal reprogramming to improve efficiency, and sometimes is required for reprogramming to occur. For example, the BAM factors (Brn2, Ascl1, and Myt1l) must work together to efficiently reprogram fibroblasts to neurons [5,6]. In other cases, additional factors may facilitate neuronal reprogramming by improving cell survival [46,103] and neuronal subtype output [104,105,106]. In all these studies, the reprogramming factors and facilitating factors are expressed under strong promoters and are expected to be at a high expression level during the reprogramming process. However, factors may interact with each other at different molecular levels and affect the “functional level” of each factor. For example, NeuroD1 has been shown to compete with Ascl1 in the determination of neuronal subtypes during forebrain development [41]. Therefore, both the expression level and functional level of the reprogramming factors would need to be considered when a combination of factors is applied. 

Another way of combining reprogramming factors is to allow for sequential expression. This can be achieved by combining two inducible expression systems (for example, Tet-on and Cumate-on) [78]. As discussed above, Sox2, as a stem cell maintenance transcription factor [29,30], can efficiently reprogram reactive glial cells to proliferative neuroblasts but generate mature neurons with less efficiency on its own [53,66]. On the other hand, NeuroD1 is a transcription factor of terminal differentiation and essential to cell cycle exit and neuron maturation. An interesting strategy would be to sequentially overexpress Sox2 and NeuroD1 using two inducible expression systems so that Sox2 can reprogram one glial cell into multiple neuroblasts, which then can be efficiently induced by NeuroD1 to become mature neurons. This way, a higher number of mature neurons would be reprogrammed per glial cell, which is hard to achieve by either factor alone. 

Lastly, transgene silencing is an important aspect of gene expression regulation, especially for long-term studies. Both genome-integrative (retroviral and lentiviral) and non-integrative (AAV) vectors can be silenced, at least partially, by the host cells through complex mechanisms that are still largely elusive [107]. Whether transgene silencing plays a significant role in controlling the expression level of the reprogramming factors during and after neuronal reprogramming would be an interesting question to address, but it is beyond the scope of this review. 

## 6. Conclusions and Future Perspectives

In sum, neuronal reprogramming, especially in vivo neuronal reprogramming, holds great promise to advance regenerative medicine. While considerable efforts have been made to identify the reprogramming factors in the past, the expression level of these factors has emerged as an important parameter for successful reprogramming. A high expression level of the reprogramming factors is usually required to break the genomic barrier of cell types. However, we believe that the dynamically controlled expression of the reprogramming factors will fine-tune the reprogramming process to improve reprogramming outcomes.

Future directions along this line of research lie in testing this strategy in different types of cell culture, as well as different injury/disease models to move towards clinical applications. Although this review focuses on the intrinsic properties of the reprogramming process, environmental cues such as injury/disease milieu are also critical to the reprogramming outcome. The hostile tissue conditions resulting from injury/disease will impose a substantial impact on neuronal reprogramming, including the survival of the reprogrammed neurons [108]. Therefore, the manipulation of local environment to improve neuronal reprogramming efficiency has been pursued in CNS injury models [15]. Furthermore, the injury mode can also influence the neuronal subtype identity of the reprogrammed neurons, as demonstrated in the retina [109]. On the other hand, the reprogrammed neurons, if substantial in number, can “reverse” the injured tissue environment back to a less hostile one [110]. Thus, both intrinsic and extrinsic factors would need to be considered when optimizing the reprogramming outcome in future investigations.

## Figures and Tables

**Figure 1 cells-13-01223-f001:**
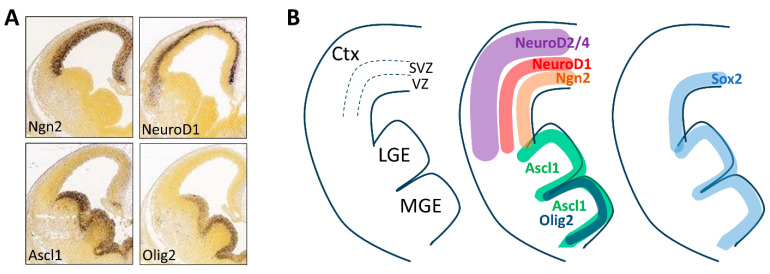
Distinct expression pattern of the reprogramming factors in the developing telencephalon. (**A**) Images from the Allen Institute for Brain Science [https://portal.brain-map.org (accessed on 24 May 2024)] showing mRNA expression of some reprogramming factors in the coronal sections of E13.5 mouse telencephalon. (**B**) Simplified schematic drawings showing anatomical structures of the E13.5 mouse telencephalon and spatial expression patterns of the reprogramming factors. Ctx, cortex; LGE, lateral ganglionic eminence; MGE, medial ganglionic eminence; VZ, ventricular zone; SVZ, sub-ventricular zone.

**Figure 2 cells-13-01223-f002:**
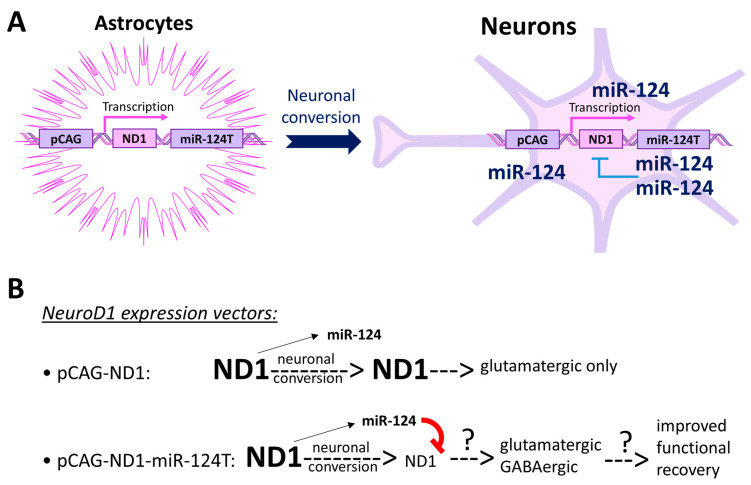
A potential vector design using miR-124 target sites to create a dynamic expression pattern of NeuroD1. (**A**) Including miR-124 target sites in the expression vector allows for the different expression levels of NeuroD1 in astrocytes and neurons. (**B**) The dynamic expression pattern of NeuroD1 may allow for the generation of neuronal subtype diversity, which could lead to improved functional recovery. ND1, NeuroD1; miR-124T, miR-124 target site.

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
