# Peer review of "Controlling the Expression Level of the Neuronal Reprogramming Factors for a Successful Reprogramming Outcome"

_cells, 2024, doi:10.3390/cells13141223_

Round 1

Reviewer 1 Report

Comments and Suggestions for Authors

This is a concise and in-depth review on neuronal reprogramming that has emerged as a potential breakthrough in regenerative medicine. The authors discussed neuronal reprogramming from a unique angle by focusing on controlling gene expression levels of the reprogramming factors, which is a topic deserving more attention. The authors also reviewed literatures for techniques that can facilitate dynamic control of gene expression and proposed interesting and sophisticated methods to control expression of reprogramming factors for a successful outcome. Given the recent debates over validation of reprogrammed neurons in vivo, this is a timely review in the neuronal reprogramming that may have a positive influence in the field for the readers of Cells.

Some comments:

1.    Page 3, Line 97: “corelate” to “correlate”

2.    Suggest changing “miR-124 binding site” to “miR-124 target site” to be consistent with the literature.

Author Response

We appreciate Reviewer 1's positive evaluation.

Comment 1: Page 3, Line 97: “corelate” to “correlate”

Response 1: This has been corrected in text.

Comment 2: Suggest changing “miR-124 binding site” to “miR-124 target site” to be consistent with the literature.

Response 2: We agree with Reviewer 1 on this comment. This has been changed accordingly in text, figure legend, and figure itself.

Reviewer 2 Report

Comments and Suggestions for Authors

Cellular reprogramming is a transformative process that alters the identity of cells, facilitating the creation of desired cell types essential for regenerative medicine. Somatic cells can be induced to adopt a desired identity through gene insertion (e.g., NeuroD1, Sox2, Ngn2, Ascl1, and microRNA), and through the application of growth factors and chemicals.

In their review, the authors provide a comprehensive summary of current approaches in cell reprogramming, emphasizing the pivotal role of gene expression regulation in achieving successful outcomes. The manuscript is commendably structured and well-written.

However, there are certain limitations that warrant attention:

1.     The specific clinical applications of cell reprogramming remain underexplored and should be thoroughly discussed. Understanding how these techniques can be effectively translated into clinical settings is crucial for their broader implementation in treating various diseases and injuries.

2.     Cell reprogramming involves converting one type of somatic cell into a different desired cell type. In many pathological conditions, such as traumatic brain and spinal cord injuries, the local microenvironment at the injury site is highly inflammatory and inhibitory to both cell reprogramming and neurogenesis. Moreover, significant cell death often occurs at the lesion site, further complicating the potential for effective repair by reprogrammed cells. The authors should address these limitations, discussing strategies to overcome these challenges and enhance the therapeutic potential of reprogrammed cells in such hostile environments.

In summary, while the review provides a valuable summary of current advancements in cell reprogramming, addressing these limitations will enrich the discussion and provide a more comprehensive view of the field's practical implications and challenges.

Comments on the Quality of English Language

the quality of English is fine.

Author Response

We appreciate Reviewer 2's positive evaluation.

Comment 1: The specific clinical applications of cell reprogramming remain underexplored and should be thoroughly discussed. Understanding how these techniques can be effectively translated into clinical settings is crucial for their broader implementation in treating various diseases and injuries.

Response 1: We agree with Reviewer 2 on this comment. We have added a sentence on Page 2, line 49 with 10 references covering various injury and disease models, in which neuronal reprogramming has been successfully demonstrated. We believe this will provide readers with resources to retrieve information on this topic and meanwhile keep the focus of this review on gene regulation of the reprogramming factors.

Comment 2: Cell reprogramming involves converting one type of somatic cell into a different desired cell type. In many pathological conditions, such as traumatic brain and spinal cord injuries, the local microenvironment at the injury site is highly inflammatory and inhibitory to both cell reprogramming and neurogenesis. Moreover, significant cell death often occurs at the lesion site, further complicating the potential for effective repair by reprogrammed cells. The authors should address these limitations, discussing strategies to overcome these challenges and enhance the therapeutic potential of reprogrammed cells in such hostile environments.

Response 2: We thank Reviewer 2 for bringing up this point. We believe this is an extremely important question to address in the entire reprogramming field before moving the technology towards clinical applications to treat different injuries/diseases. We have added a paragraph on Page 9, line 378 to discuss this point as future perspectives.

Reviewer 3 Report

Comments and Suggestions for Authors

The manuscript from Mseis-Jackson discusses an interesting yet not enough explored aspect of direct neuronal reprogramming, namely to which extent the expression of reprogramming factors can affect the conversion itself, in terms of efficiency but also maturation. After briefly revising the expression
pattern of some reprogramming factors during embryonic neurogenesis, the authors discuss the contradictory results obtained upon the expression of different levels of NeuroDl in microglia, and propose ways to control it, including the use of druggable promoters and miRNA-mediated detargeting.

The manuscript is overall well-written and nice to read it.

I have few suggestions to improve the manuscript:

_ the authors mentioned mainly transcription factors associated to a glutamatergic fate (Ngn2, NeuroDl). However, a great part of the manuscript focuses on NeuroDl. It would be better for the reader to know it, either by mentioning in the title or in the abstract;

_ about the importance of the expression level: it would be worth mentioning a recent paper (Hersbach et al., 2022: PMID: 36106915 ), where it was shown that the effect of the reprogramming factor is not proportional to its expression, but rather biphasic. This is a rather important point to discuss, as the untold assumption in the reprogramming field is that the higher the expression, the better the conversion.

_ likewise , when the authors discuss the use of the TetON system , it would be worth mentioning that also hepatocytes (PMID:21962918 ) and astrocytes (PMID : 38956165 ) have been successfully reprogrammed using the TetON system, thus making it a rather universal system for direct neuronal conversion.

_ the authors mentioned Sox2 as neuronal reprogramming factor: technically speaking, this is not true, as in the cited papers (53,54) the authors clearly state that the forced expression of Sox2 induces a neuroblast identity (Dcx+), that then will further lead to generate neurons. Please rephrase the sentence.

Author Response

We appreciate Reviewer 3's positive evaluation.

Comment 1: the authors mentioned mainly transcription factors associated to a glutamatergic fate (Ngn2, NeuroDl). However, a great part of the manuscript focuses on NeuroDl. It would be better for the reader to know it, either by mentioning in the title or in the abstract.

Response 1: We agree with Reviewer 3 on this comment. We now have mentioned this in the Abstract on Page 1, line 19.

Comment 2: about the importance of the expression level: it would be worth mentioning a recent paper (Hersbach et al., 2022: PMID: 36106915 ), where it was shown that the effect of the reprogramming factor is not proportional to its expression, but rather biphasic. This is a rather important point to discuss, as the untold assumption in the reprogramming field is that the higher the expression, the better the conversion.

Response 2: We agree with Reviewer 3 on this comment. We have added a section on Page 5, line 202 to discuss this and added this reference.

Comment 3: likewise , when the authors discuss the use of the TetON system , it would be worth mentioning that also hepatocytes (PMID:21962918 ) and astrocytes (PMID : 38956165 ) have been successfully reprogrammed using the TetON system, thus making it a rather universal system for direct neuronal conversion.

Response 3: We thank Reviewer 3 for this comment. We have added this information on Page 7, line 293 to reflect this point with references.

Comment 4: the authors mentioned Sox2 as neuronal reprogramming factor: technically speaking, this is not true, as in the cited papers (53,54) the authors clearly state that the forced expression of Sox2 induces a neuroblast identity (Dcx+), that then will further lead to generate neurons. Please rephrase the sentence.

Response 4: We thank Reviewer 3 for pointing this out. We now have rephrased the sentence on Page 5, line 239.